# The pesticide chlorpyrifos promotes obesity by inhibiting diet-induced thermogenesis in brown adipose tissue

Bo Wang[1,2,3], Evangelia E. Tsakiridis[1,2], Shuman Zhang[1,2], Andrea Llanos[1,4], Eric M. Desjardins[1,2], Julian M. Yabut[1,2], Alexander E. Green [1,2], Emily A. Day [1,2], Brennan K. Smith[1,2], James S. V. Lally[1,2], Jianhan Wu[1,2], Amogelang R. Raphenya [1,5], Krishna A. Srinivasan[1,5], Andrew G. McArthur [1,5], Shingo Kajimura [6], Jagdish Suresh Patel [7,8], Michael G. Wade[9], Katherine M. Morrison [1,10], Alison C. Holloway [1,4] & Gregory R. Steinberg [1,2,5✉]

Obesity results from a caloric imbalance between energy intake, absorption and expenditure. In both rodents and humans, diet-induced thermogenesis contributes to energy expenditure and involves the activation of brown adipose tissue (BAT). We hypothesize that environmental toxicants commonly used as food additives or pesticides might reduce BAT thermogenesis through suppression of uncoupling protein 1 (UCP1) and this may contribute to the development of obesity. Using a step-wise screening approach, we discover that the organophosphate insecticide chlorpyrifos suppresses UCP1 and mitochondrial respiration in BAT at concentrations as low as 1 pM. In mice housed at thermoneutrality and fed a high-fat diet, chlorpyrifos impairs BAT mitochondrial function and diet-induced thermogenesis, promoting greater obesity, non-alcoholic fatty liver disease (NAFLD) and insulin resistance. This is associated with reductions in cAMP; activation of p38MAPK and AMPK; protein kinases critical for maintaining UCP1 and mitophagy, respectively in BAT. These data indicate that the commonly used pesticide chlorpyrifos, suppresses diet-induced thermogenesis and the activation of BAT, suggesting its use may contribute to the obesity epidemic.

[1] Centre for Metabolism, Obesity and Diabetes Research, McMaster University, Hamilton, ON, Canada. [2] Division of Endocrinology and Metabolism, Department of Medicine, McMaster University, Hamilton, ON, Canada. [3] State Key Laboratory of Animal Nutrition, College of Animal Science and Technology, China Agricultural University, Beijing, PR China. [4] Department of Obstetrics and Gynecology, McMaster University, Hamilton, ON, Canada. [5] Department of Biochemistry and Biomedical Sciences, McMaster University, Hamilton, ON, Canada. [6] Beth Israel Deaconess Medical Center and Harvard Medical School, Boston, MA, USA. [7] Institute for Modeling Collaboration and Innovation, University of Idaho, Moscow, ID, USA. [8] Department of Biological Sciences, University of Idaho, Moscow, ID, USA. [9] Environmental Health Science & Research Bureau, Health Canada, Ottawa, ON, Canada. [10] Department of Pediatrics, McMaster University, Hamilton, ON, Canada. ✉email: gsteinberg@mcmaster.ca

Obesity is a major risk factor for type 2 diabetes (T2D), non-alcoholic fatty liver disease (NAFLD), and cardiovascular disease[1] that arises from a caloric surplus of as little as 10–30 kcal per day[2]. And while increased consumption of energy-dense foods and reduced physical activity are commonly thought to be the major contributors to this caloric imbalance[1], diet-induced thermogenesis is a quantitatively important component of the energy balance equation[3]. In adult humans, recent studies have indicated that diet-induced thermogenesis requires the activation of brown adipose tissue (BAT)[4–6], however, the determinants regulating this process and why they may differ between individuals are not fully understood.x

A key protein regulating BAT thermogenesis is uncoupling protein 1 (UCP1). Genetic removal of *Ucp1* in mice promotes obesity and insulin resistance when mice are fed a diet high in fat and housed under thermoneutral conditions (29–30 °C), indicating a vital role for this protein in diet-induced thermogenesis[7]. UCP1 is also important for diet-induced thermogenesis in humans, as a genetic loss of function mutation reduces postprandial thermogenesis in response to a high-fat meal[8]. While genetic polymorphisms in *Ucp1* have been linked to obesity in some studies[9–11], this has not been observed in large genome-wide association studies[12], suggesting that other factors besides genomic alterations in *Ucp1* may be contributing to reduced BAT activity in obese humans[13].

Over the last two decades, a number of environmental toxicants have been linked to the development of obesity through their effects on gut microbiota[14], energy intake, or adipogenesis[15–18] (for review see refs. [19,20]), however, only a few of these studies have examined a role for these agents to contribute to obesity by inhibiting energy expenditure[21–23] and/or BAT thermogenesis[24,25] (reviewed in refs. [18,20]). We hypothesized that environmental toxicants commonly present in food might reduce diet-induced thermogenesis through suppression of UCP1, the defining protein of human BAT thermogenesis[26–31]. Through a step-wise screening of several common food contaminants or additives, we discovered that the organophosphate insecticide chlorpyrifos (CPF) potently suppressed the expression of UCP1 and mitochondrial respiration in brown adipocytes at concentrations as low as 1 pM. Chloropyrifos-induced suppression of diet-induced thermogenesis in BAT has also been observed in mice fed a diet high in fat and housed at thermoneutrality where it promoted greater obesity, NAFLD, and insulin resistance. Reductions in BAT thermogenesis by CPF were associated with reductions in cAMP and protein kinases critical for regulating UCP1 and mitophagy. These data indicate that the commonly used pesticide CPF, at very low concentrations, suppresses the activation of BAT, suggesting that its use may contribute to the obesity epidemic.

## Results

We examined the effects of 34 chemicals—selected based on their widespread presence in food due to agriculture practices, food processing, and/or packaging (Table 1) on the expression of *Ucp1* in immortalized brown adipocytes generated from *Ucp1*-luciferase reporter mice[32]. Of the 34 chemicals tested, only the pesticide CPF significantly decreased *Ucp1* promoter activity and mRNA expression by at least 25% across six orders of magnitude of concentrations tested down to 1 pM (Fig. 1a, b, Supplementary Fig. 1a–c).

CPF is an organophosphate insecticide with broad-spectrum activity against many foliar and soil insects and is commonly used for pest control in a wide range of field crops and fruit[33]. Previous studies in animals have found that high doses of CPF, which inhibits plasma butyryl cholinesterase and brain acetylcholinesterase activity, can promote obesity and glucose dysregulation through mechanisms that are poorly defined but may involve increases in energy intake and/or alterations in the gut microbiome[34–39]. However, whether doses of CPF below those that inhibit acetylcholinesterase and closer to human-relevant exposures promote obesity in rodents or humans is currently unknown. We found that consistent with the acute suppression of *Ucp1* promoter activity and *Ucp1* mRNA, chronic CPF (6 days) also suppressed UCP1 protein in brown adipocytes at a concentration of 1 pM (Fig. 1c and Supplementary Fig. 1c). To further characterize the effects of CPF, we completed global RNA-sequencing after just 4 h of CPF treatment and found that RNA transcripts related to mitochondrial function/metabolism were most affected (Fig. 1d). Notably, in addition to *Ucp1*, RNA transcripts are critical for regulating fatty acid oxidation (*Cpt1a*, *Cpt1b*, *Acat3*), and Cytochrome C oxidase assembly factor (*Cox16*) (Fig. 1e) were reduced with CPF treatment. Consistent with previous studies in UCP1 null mice[40,41], compensatory upregulation of some mitochondrial transcripts was also observed (Fig. 1e). Importantly, consistent with alterations in transcriptional programs regulating mitochondrial function, chronic treatment with CPF-lowered respiration in the presence of the mitochondrial ATP synthase inhibitor oligomycin and the mitochondrial uncoupler FCCP; indicating impaired mitochondrial leak and maximal respiration, respectively (Fig. 1f). CPF also lowered cytochrome *c* oxidase activity (Fig. 1g) and mitochondrial membrane potential (Fig. 1h). These reductions in mitochondrial function were not due to impairments in BAT differentiation as Oil Red O staining and key transcription factors/markers of BAT differentiation (*Prdm16*, *Ppargc1a*, *Pparg*, *Cidea*) were not altered by CPF (Supplementary Fig. 1d, e). Impaired mitochondrial function was not observed following acute CPF exposure (4 h) (Supplementary Fig. 1f, g) as had previously been observed at high concentrations of CPF (40 μM)[42]. These data indicate that concentrations of CPF significantly below those causing toxicity in animal studies and more aligned with real-world human exposures[43] inhibit transcriptional programs crucial for regulating BAT mitochondrial function.

CPF is readily detected in a wide range of foods including fruits, vegetables, grains, beans, nuts, legumes, dairy, meat, fish,

**Table 1 List of tested chemicals.**

| Pesticides | Pesticide metabolites | Food-packaging compounds | Food processing compounds |
|---|---|---|---|
| 2,4-D | AMPA | BPA | Zearalanone |
| Atrazine | Atrazine mercapturate | BPAF | Allura red |
| Chlorpyrifos | 6-Chloronicotinic acid | BPB | Brilliant blue |
| Deltamethrin | Chlorpyrifos oxon | BPS | Cresidinesulfonic acid |
| Glyphosate | Desethyl atrazine | BPF | Sulfanilic acid |
| Imidacloprid | Diaminochlorotriazine | BPA-b-D-glucuronide | Sunset yellow |
| Metam sodium | MITC | MINCH | Tartrazine yellow |
| Permethrin | 3-Phenoxybenzoic acid | PFOA | |
| S-Metolachlor | TCP-y | TBPA | |

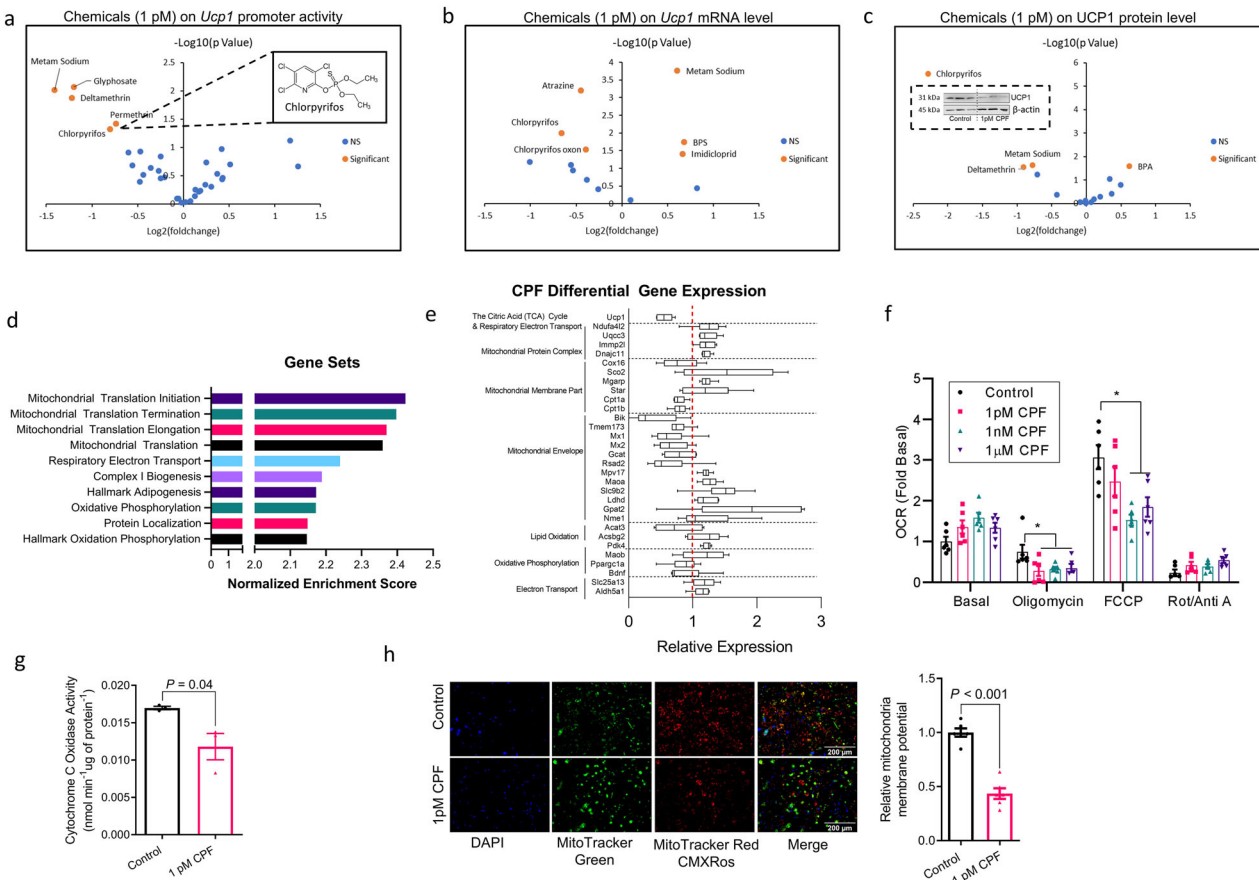

**Fig. 1 CPF inhibits the thermogenic gene program and mitochondrial respiration in cultured brown adipocytes. a** *Ucp1* promoter activity of brown adipocytes treated with 34 different chemical compounds at the dose of 1 pM for 16 h. **b** *Ucp1* mRNA levels of brown adipocytes treated with compounds at 1 pM for 4 h. **c** UCP1 protein concentration in brown adipocytes treated with compounds at 1 pM for 6 days. **d** and **e** Gene Set Enrichment Analysis of mRNA-sequencing data (**d**) of brown adipocytes treated with 1 pM chlorpyrifos (CPF) for 4 h and relative expression of genes within gene sets (**e**, Whiskers: min to max). **f** Oxygen consumption rate of brown adipocytes treated with different doses of CPF for 6 days. **g** Cytochrome C oxidase activity of brown adipocytes treated with 1 pM CPF for 6 days. **h** Representative images of mitochondria stained by MitoTracker Green and quantification of mitochondria potential reflected by MitoTracker Red CMXRos in brown adipocytes treated with 1 pM CPF for 6 days. Significant differences between 3 or more mean values were determined by one-way ANOVA with the post hoc Bonferroni's multiple comparisons test; differences between 2 mean values were determined by a two-tailed Student's $t$-test. Data presented are mean ± SEM, $n = 6$, *$p < 0.05$. Scale bar = 200 μm.

and eggs, at concentrations ranging from 0.002 to 436 μg/kg[44]. We examined the effects of CPF in male C57BL6J mice that were fed a control diet (CD, 10 kcal% fat) or high-fat diet (HFD, 45 kcal% fat) supplemented with two doses of CPF (5 or 20 mg/kg diet), which equated to an exposure of ~0.5 or 2 mg/kg body weight in mice fed a control diet housed at room temperature (CD-RT) with slightly lower exposure in HFD fed mice housed at thermoneutrality (TN, Supplementary Fig. 2a). In contrast to an acute gavage of CPF (100 mg/kg) which resulted in dramatic inhibition of ChE activity in the serum, we found that CPF in the diet did not inhibit acetyl-cholinesterase activity in skeletal muscle (Supplementary Fig. 2b and c), which is consistent to a previous study[34].

Mice housed at room temperature (RT, 21–23 °C), common in most animal vivarium, are under cold stress and have increased sympathetic tone that upregulates basal metabolic rate through adaptive thermogenesis[7,45]. This elevation in basal metabolic rate masks the effects of diet-induced thermogenesis mediated through β-adrenergic driven activation of BAT UCP1 compared to housing mice at thermoneutrality (TN, 29–30 °C) as evidenced by the development of obesity in mice lacking UCP1 at TN but not RT. Thus TN housing is a temperature range for mice that

better mimics a condition in which humans reside[45] and as such recent studies have indicated that it more accurately models the development of metabolic diseases including obesity, insulin resistance, and NAFLD[46,47]. Therefore, mice were housed at either RT or TN. To the best of our knowledge, this is the first time an environmental toxicant has been tested in rodents housed at TN.

On a control diet, CPF supplementation did not influence body weight (Supplementary Fig. 3a, b), fat mass (Supplementary Fig. 3c, d), glucose tolerance (Supplementary Fig. 3e, f), or insulin sensitivity (Supplementary Fig. 3g, h) in RT or TN conditions. However, when mice were housed at TN and fed an HFD, CPF treatment increased body mass (Fig. 2a), adiposity (Fig. 2b), and reduced glucose tolerance (Fig. 2c), and insulin sensitivity (Fig. 2d) compared to non-CPF treated mice. Importantly, these effects were blunted in mice housed at RT such that only the highest dose of CPF (2.0 mg/kg) promoted weight gain, glucose intolerance, and insulin resistance (Supplementary Fig. 4a–e). Consistent with increased adiposity under TN-HFD-fed conditions (Fig. 2b, e), CPF increased the size of WAT lipid droplets (Fig. 2f) and elevated serum free fatty acids (FFA, Fig. 2g) and triglycerides (TGs, Fig. 2h). Consistent with increased obesity, insulin resistance and FFAs HFD-fed mice treated with CPF and

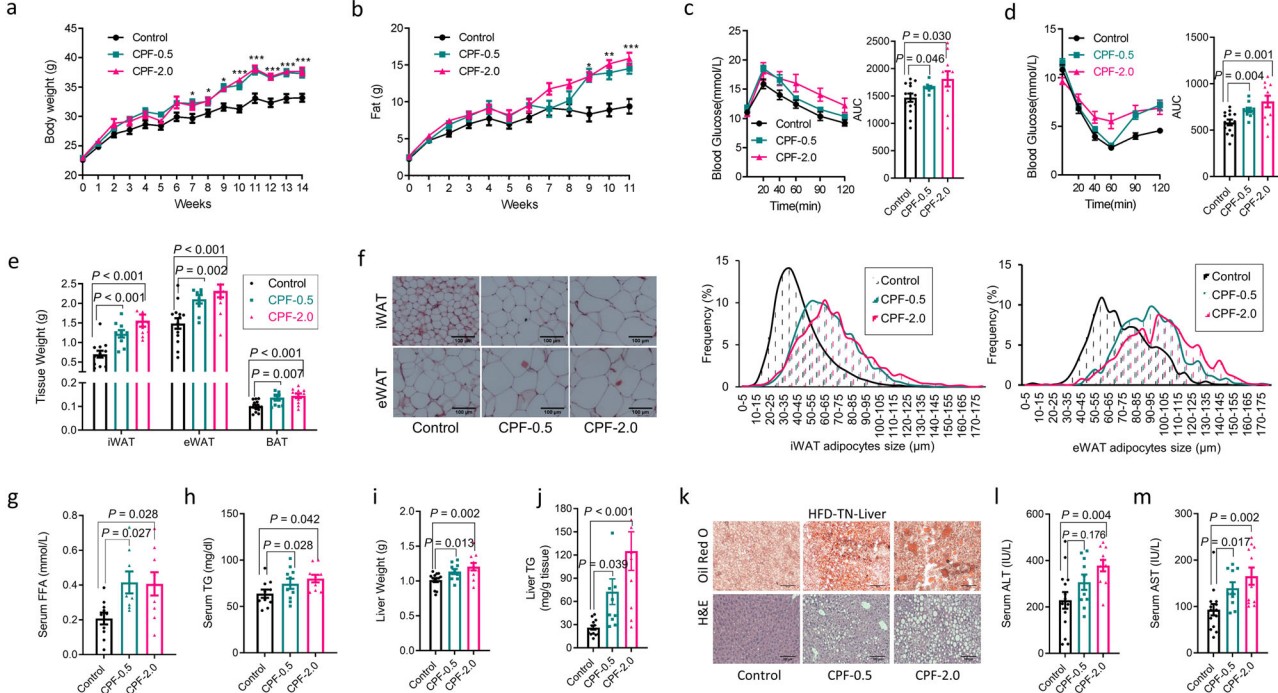

**Fig. 2 CPF promotes obesity and metabolic dysfunction in mice fed a high-fat diet and at thermoneutrality.** C57BL/6J male mice were fed a HFD (45 kcal% fat) supplemented with 0 (Control), 0.5 mg/kg/BW (CPF-0.5) or 2.0 mg/kg/BW (CPF-2.0) chlorpyrifos and housed at thermoneutrality (TN, 30 °C) for 14 weeks. **a** Bodyweight. **b** Fat mass. **c** Glucose tolerance test. **d** Insulin tolerance test (ITT). **e** Adipose tissues weight. **f** Representative images of H&E stained inguinal white adipose tissue (iWAT) and epididymal white adipose tissue (eWAT). **g** Serum-free fatty acid (FFA) level. **h** Triacylglycerol (TG) level. **i** Liver weight. **j** Liver TG concentration. **k** Representative images of Oil Red O-stained liver lipids and H&E-stained liver. **l** Serum alanine aminotransferase (ALT) level. **m** Aspartate aminotransferase (AST) level. Significant differences between mean values were determined by one-way ANOVA with the post hoc Bonferroni's multiple comparisons test. Data presented are mean ± SEM, n = 10, *p < 0.05, **p < 0.01, ***p < 0.001. Scale bar = 100 μm.

housed at TN had increased liver weights (Fig. 2i), liver TGs (Fig. 2j, k) and serum ALT (Fig. 2l) and AST (Fig. 2m) indicative of greater NAFLD than HFD controls. Importantly, these effects of CPF were blunted when fed an HFD at RT and were not observed when mice were fed with CD indicating these markers of NAFLD/liver toxicity were likely secondary to the increased weight rather than toxicity to CPF (Supplementary Figs. 5 and 6, respectively). These data indicate that low concentrations of CPF, which do not inhibit AchE activity, when combined with TN housing and an HFD promote obesity, insulin resistance, and NAFLD.

To investigate the mechanisms by which CPF promoted HFD-induced weight gain, caloric intake and energy expenditure were assessed in metabolic cages. Consistent with previous studies mice housed at TN had lower caloric intake compared to mice housed at RT[48] and regardless of housing temperature, CPF increased caloric intake[36,49] (Fig. 3a). No differences in physical activity were observed between groups (Fig. 3b). As anticipated, TN lowered absolute (not corrected for body mass) resting oxygen consumption compared to RT housing (Fig. 3c). Importantly, CPF suppressed oxygen consumption by ~15% in mice housed at TN but not RT (Fig. 3c). These data indicate that CPF suppresses energy expenditure at TN but not RT. To further examine whether reductions in energy expenditure were important for the weight gain observed in HFD-fed mice housed at TN, in separate experiments mice were fed with CPF and allowed ad libitum access to food or were pair-fed to controls (Fig. 3d). Importantly, even in the pair-fed group CPF treated mice had a strong tendency for higher body weight and increased adiposity after 9 weeks of HFD feeding when the amount of food intake was matched with that of the control mice (Fig. 3e and f). Collectively

these data suggest that CPF reduces energy expenditure, contributing to the increased weight gain observed at TN.

Our data demonstrating reductions in oxygen consumption at TN, but not at RT with CPF, are similar to observations in UCP1 null mice, which also only develop HFD-induced obesity when housed at TN; an effect attributed to impairments in BAT diet-induced thermogenesis[7], We found that in HFD-fed mice housed at TN, CPF reduced *Ucp1* mRNA and protein expression (Fig. 4a and b) and increased the size and number of lipid droplets (Fig. 4c) within the BAT. In addition to suppressing UCP1, transmission electron microscopy revealed an increased abundance of large mitochondria (Fig. 4d and e) with disrupted cristae (Fig. 4d and f). These effects of CPF on UCP1 and BAT morphology and mitochondria were not evident when mice were housed at RT (Supplementary Fig. 7). These data indicate that consistent with increased weight gain and metabolic dysfunction when mice are housed at TN, CPF reduces BAT UCP1 expression and mitochondrial morphology.

To directly test the hypothesis that CPF was suppressing diet-induced thermogenesis we conducted a subsequent study in which mice were housed at TN and fed an HFD with or without CPF for 5 weeks (Fig. 5a), a time point preceding any differences in body mass or adiposity (Supplementary Fig. 8a, b). Mice were then placed in metabolic cages and fasted overnight before refeeding. CPF-treated mice had similar oxygen consumption (Fig. 5b) and energy expenditure (Fig. 5c) during fasting. However, CPF-treated mice had an ~10% reduction in oxygen consumption and energy expenditure during refeeding (Fig. 5b and c) indicative of impaired diet-induced thermogenesis. Importantly, this was not due to differences in food intake during the refeed (Fig. 5d) or activity levels (Fig. 5e). These

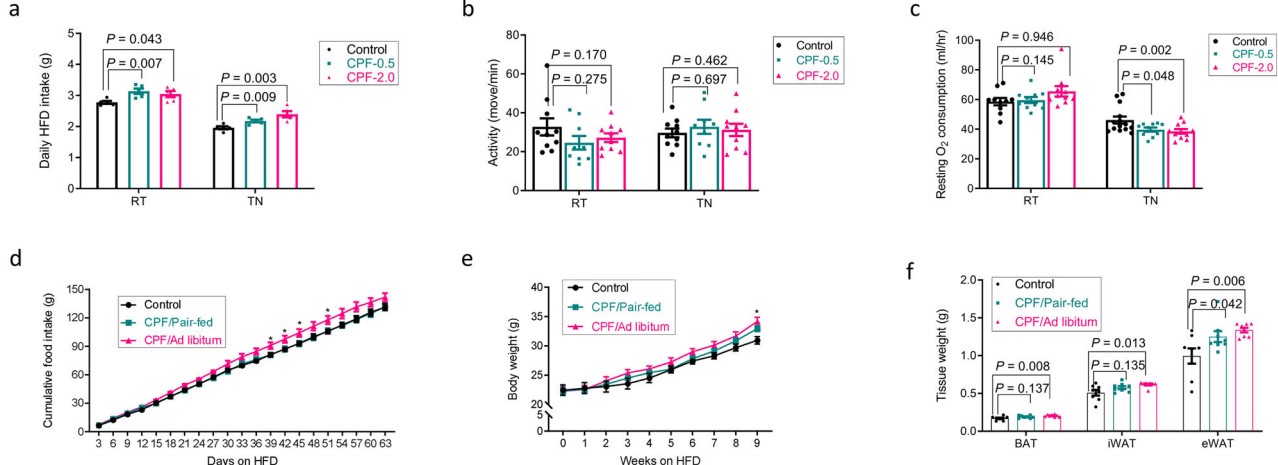

**Fig. 3 Dietary exposure to chlorpyrifos promotes weight gain in HFD-fed mice housed at thermoneutrality even in the absence of increased food intake. a–c** C57BL/6J male mice were fed a HFD (45 kcal% fat) supplemented with 0 (Control), 0.5 mg/kg/BW (CPF-0.5) or 2.0 mg/kg/BW (CPF-2.0) chlorpyrifos and housed at thermoneutrality (TN, 30 °C) for 14 weeks. Daily food intake (**a**) was recorded weekly; animal activity (**b**) and resting $O_2$ consumption rate (**c**) were measured in CLAMS; $n = 10$. **d–f** C57BL/6J male mice were treated with HFD (45 kcal% fat) supplemented with 0 (Control) or 2.0 mg/kg BW CPF for 9 weeks, the food provided to one group of the CPF mice (CPF/Pair-fed) was matched with that of the control group. The food intake ($n = 4$, **d**), body weight ($n = 8$, **e**), and fat tissue weight ($n = 8$, **f**) were measured. Significant differences between mean values were determined by one-way ANOVA with the post hoc Bonferroni's multiple comparisons test. Data presented are mean ± SEM, *$p < 0.05$.

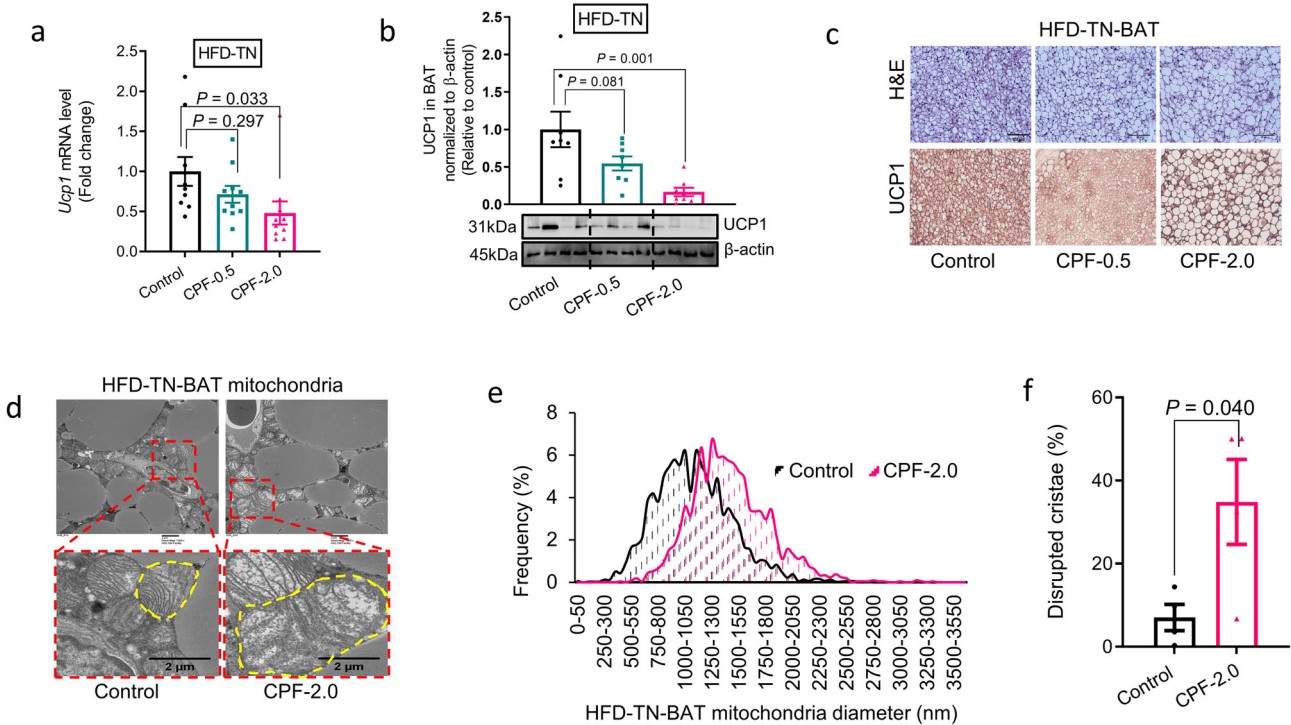

**Fig. 4 Dietary exposure to chlorpyrifos impairs mitochondrial function and thermogenesis in brown adipose tissue in vivo.** C57BL/6J male mice were fed a HFD (45 kcal% fat) supplemented with 0 (Control), 0.5 mg/kg/BW (CPF-0.5), or 2.0 mg/kg/BW (CPF-2.0) chlorpyrifos and housed at thermoneutrality (TN, 30 °C) for 14 weeks. **a** mRNA expression level of *Ucp1* in BAT, $n = 10$. **b** UCP1 protein in BAT of mice at TN, $n = 8$. **c** Representative immunohistochemistry images showing UCP1 in BAT and H&E images, Scale bar = 100 μm. **d** Representative electron micrographs for mitochondria, scale bar = 2 μm. **e** Diameter distribution of mitochondria. **f** Quantification of disrupted mitochondria cristae, $n = 4$. Significant differences between 3 or more mean values were determined by one-way ANOVA with the post hoc Bonferroni's multiple comparisons test; differences between 2 mean values were determined by a two-tailed Student's *t*-test. Data presented are mean ± SEM.

data suggest that CPF reduces energy expenditure by suppressing diet-induced thermogenesis.

In mice diet-induced thermogenesis is mediated through nor-epinephrine stimulated β3-adrenergic receptor (β3-AR) activation, which results in activation of adenylyl cyclase, and subsequent increases in cAMP and PKA activity[50,51]. To examine whether this pathway may be altered with CPF, we measured cAMP in response to the pan beta-agonist isoproterenol and found that CPF reduced

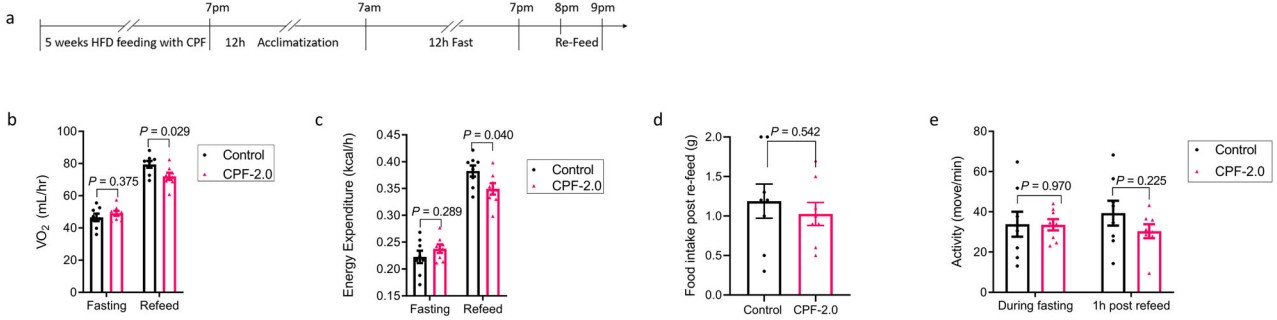

**Fig. 5 Chlorpyrifos inhibits diet-induced thermogenesis.** C57BL/6J male mice were treated with an HFD (45 kcal% fat) supplemented with 0 (control) or 2.0 mg/kg/BW (CPF-2.0) chlorpyrifos for 5 weeks, followed by 12 h fasting and 2 h-refeeding. **a** A timeline showing treatment procedure. **b** Oxygen consumption. **c** Energy expenditure. **d** Food intake during 1 h refeed. **e** Physical activity levels from 6–7 p.m. (fasting) and from 7–8 p.m. (refeed). $n = 10$. Significant differences between mean values were determined by a two-tailed Student's $t$-test. Data presented are mean ± SEM.

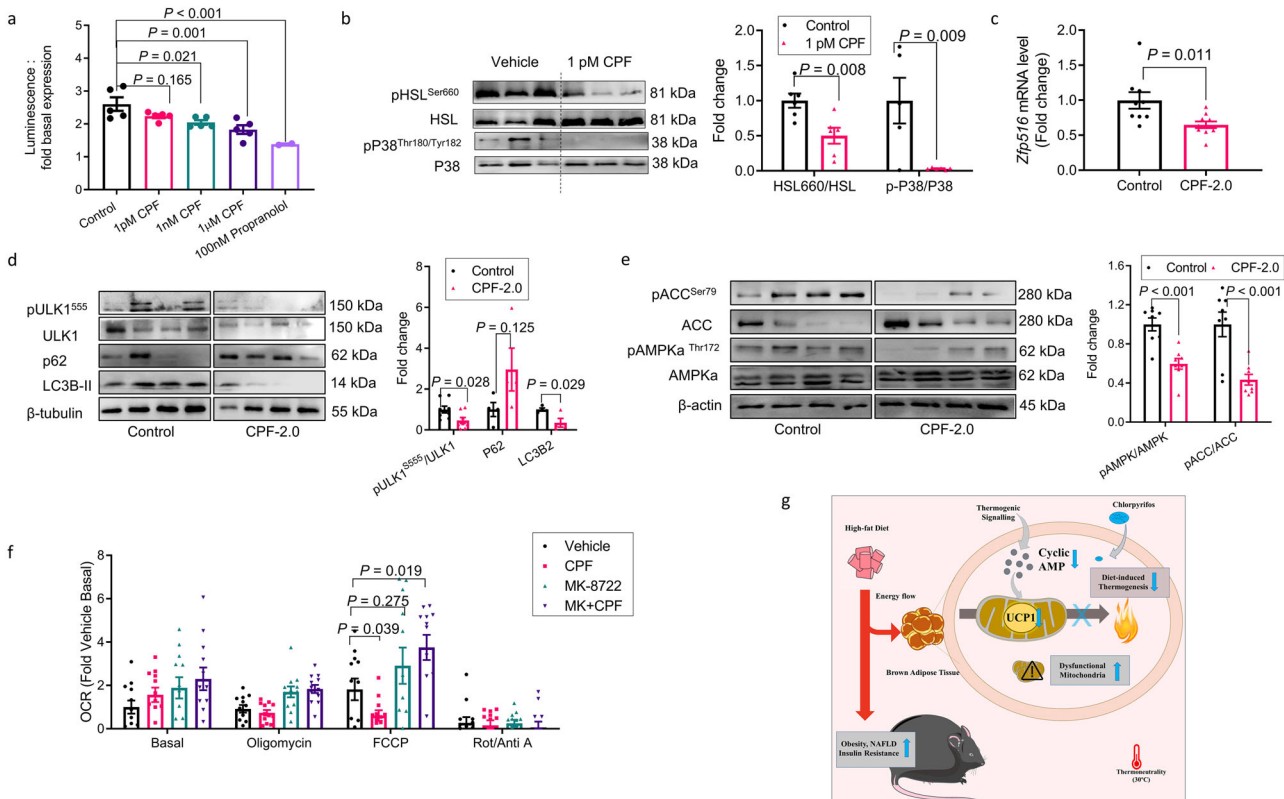

**Fig. 6 Chlorpyrifos inhibits diet-induced thermogenesis and β-adrenergic signaling in brown adipose tissue. a** Mature brown adipocyte treated with vehicle or chlorpyrifos (CPF) or propranolol for 7 days. Fold change of cAMP levels in response to 30 min isoproterenol stimulations, $n = 5$. **b–e** Phosphorylation of HSL and p38 MAPK (**b**, $n = 6$); *Zfp516* mRNA expression (**c**, $n = 10$), pULK1[555], ULK, p62, LC3B-II, pACC, ACC, pAMPKa[Thr172] and AMPK proteins (**d**, **e**, $n = 8$) in BAT of C57BL/6J mice fed a HFD with 0 (control) or 2.0 mg/kg/BW (CPF-2.0) CPF and housed at thermoneutrality (TN, 30 °C) for 14weeks. **f** Oxygen consumption rate of brown adipocytes treated with MK-8722 and/or CPF for 6 days, $n = 12$. **g** Graphical abstract showing CPF induced obesity, non-alcoholic fatty liver disease (NAFLD), and insulin resistance by inhibiting diet-induced thermogenesis. Significant differences between three or more mean values were determined by one-way ANOVA with the post hoc Bonferroni's multiple comparisons test; differences between two mean values were determined by a two-tailed Student's $t$-test. Data presented are mean ± SEM.

cAMP in isolated brown adipocytes (Fig. 6a). To examine potential mechanisms which might be contributing to this reduction in cAMP several experiments were completed. We first hypothesized that CPF may directly inhibit the β₃-AR. We built a homology model of the β₃-AR which was used as a starting structure to carry out classical molecular dynamic simulations (MD). Snapshots obtained from MD simulation were then used to dock CPF. The most populated docked pose shows that CPF is predicted to bind to the known pocket in the β₃-AR and predominantly forms hydrophobic contacts and hydrogen

bonds with S200. (Supplementary Fig. 9a) Surprisingly, the docking score of the most populated CPF docking pose was higher (−7.4 kcal/mol, lower the better) than cyanopindolol (−8.8 kcal/mol), a potent β-adrenergic receptor antagonist, suggesting a weaker binding. To validate the modeling observations, studies using a radioligand competition assay of CPF or its metabolite CPF-oxon with cyano-pindolol were conducted. However, in contrast to the docking studies, we found that neither CPF nor CPF-oxon competed with cyanopindolol for β₃-AR binding (Supplemental Fig. 9b). These data

suggest that neither CPF nor its metabolite CPF-oxon directly bind the β₃-AR.

We subsequently interrogated the hypothesis that CPF could be reducing cAMP by altering the expression of β-ARs (Adrb), adenylyl cyclase (Adcy), or phosphodiesterases (Pde) but found no change in isolated BAT cells (Supplementary Fig. 9c–e). In addition to direct binding or alterations in expression, cAMP production is dependent on membrane localization of adenylate cyclase therefore we subsequently examined whether this might be reduced by CPF treatment. Interestingly, CPF lead to a reduction of membrane-bound ADCY3 with a corresponding increase in cytoplasmic localization (Supplementary Fig. 9f). These data suggest that CPF may reduce cAMP by reducing membrane localization, however, the exact mechanisms that contribute to this effect require further investigation.

To examine the downstream consequences of this reduction in cAMP we measured the phosphorylation and activation of hormone sensitive lipase (HSL), p38MAPK, and AMP-activated protein kinase (AMPK), which are important for increasing lipolysis, UCP1 expression, and mitophagy respectively[50–53]. Reductions in cAMP in isolated brown adipocytes (Fig. 6a) were associated with reduced activating phosphorylation of HSL (Ser660) and p38MAPK (Fig. 6b); effects similar to the pan-beta antagonist propranolol (Supplementary Fig. 10a). Reductions in the phosphorylation of PKA, HSL, and p38 MAPK were also observed in BAT of CPF-treated mice fed an HFD and housed at TN (Supplementary Fig. 10b–d). Concomitant with these reductions in cAMP the expression of Zfp516, a cAMP-responsive transcription factor regulating UCP1[54], was also reduced (Fig. 6c).

Treatment of mouse and human BAT cells with norepinephrine increases cAMP and AMPK activity[55]. Similar to CPF-treated mice, mice lacking AMPK in BAT have an increased abundance of large mitochondria with disrupted cristae, an effect which is mediated through AMPK-induced increases in mitophagy[55]. We found that consistent with mice genetically lacking AMPK in BAT, CPF-treated mice and BAT cells have reduced activating phosphorylation of AMPK and ULK1 and reduced accumulation of LC3B-II and increased p62, the latter being two markers of reduced autophagic flux (Fig. 6d and Supplementary Fig. 10e). There was also reduced phosphorylation of ACC, another downstream substrate of AMPK important for regulating fatty acid oxidation and fatty acid synthesis (Fig. 6e and Supplementary Fig. 10e). To test whether this inhibition of AMPK was important for mediating the inhibitory effects of CPF on mitochondrial function, brown adipocytes were treated with the highly selective AMPK agonist MK-8722[56] which eliminated the ability of CPF to suppress mitochondrial uncoupling by FCCP (Fig. 6f). These data indicate that CPF reduces BAT cAMP and downstream signaling leading to reductions in UCP1 and pathways critical for controlling mitochondrial function.

## Discussion
A small imbalance between energy expenditure and energy intake/absorption increases adiposity and over time can lead to the development of obesity. Due to the lipophilicity of environmental toxicants, adipose tissue is the major site of environmental toxicant accumulation[57,58]. In agreement with previous studies[35–38], we found that treatment of HFD-fed mice with 2.0 mg/kg body weight of CPF increased caloric intake and promoted obesity, insulin resistance, and NAFLD when mice were housed at RT. Interestingly no effect of this dose was observed on metabolic parameters when this dose of CPF was combined with the control low-fat diet indicating important interactions between CPF and dietary fat. Importantly, this dose of CPF is below that associated with the inhibition of acetyl-

cholinesterase activity or toxicity and is more aligned with real-world exposures in humans[39,59].

When mice are housed at TN, but not RT, a low concentration of CPF (0.5 mg/kg body weight) promotes weight gain, insulin resistance, and NAFLD. To the best of our knowledge, this is the first toxicological study conducted in rodents that have been completed under TN conditions. The findings demonstrating effects of low concentrations of CPF at TN, but not RT, are consistent with recent studies that have highlighted that RT does not allow for accurate modeling of human disease due to increased thermogenic demands that mask biological processes that occur in humans who predominately reside at TN[60]. A key reason for this is because at RT mice require adaptive thermogenesis to maintain their body temperature which in turn masks diet-induced thermogenesis. Therefore, the observation of increased weight gain at TN but not RT, despite increases in caloric consumption under both housing conditions, suggested that low dose CPF may be inhibiting diet thermogenesis and this could be critical for mediating the increased weight gain. Consistent with this hypothesis pair-feeding experiments indicated that CPF mice housed at TN who were calorically matched to the lower food intake of controls still gained more weight suggesting reductions in energy expenditure were important. To further examine whether diet-induced thermogenesis was critical for the weight gain mice were housed at TN and fed an HFD containing CPF for 4 weeks, and before any differences in adiposity emerged, were fasted and refed in metabolic cages. These experiments demonstrated that despite similar activity levels and food consumption, oxygen consumption was reduced in CPF-treated mice during refeeding. These findings indicating that CPF suppresses diet-induced thermogenesis indicate a mechanism by which CPF promotes obesity and metabolic dysfunction. They also suggest that since to date all environmental toxicants have been tested in rodents housed at RT, the effects of these chemicals on human health may have been underestimated and therefore additional toxicological testing at TN may be important.

BAT and UCP1 are important for diet-induced thermogenesis in both mice and humans[50]. Increased weight gain and metabolic dysfunction in mice treated with low doses of CPF housed at TN were associated with a reduction in UCP1 and alterations in BAT mitochondria morphology. Interestingly, this effect was not observed when mice are housed at RT potentially because of the increased adrenergic drive which may have overridden the suppressive effects of CPF. These findings are in agreement with our observations in vitro that low pM concentrations of CPF suppress cAMP and downstream signaling pathways in BAT cells, effects associated with reductions in UCP1 expression, mitochondrial function, and oxygen consumption. In rodents, diet-induced thermogenesis is dependent on norepinephrine-induced activation of the β3-AR and UCP1 and in the current study, CPF exposure led to reductions in both cAMP and UCP1. We hypothesized that this may be due to binding of CPF to the β3-AR or reductions in β3-AR expression levels, however, we found no effect of CPF on either of these parameters. Previous studies in developing neurons found that CPF may reduce the activity of adenyl cyclase[61] and consistent with this observation we observed a reduction in membrane localization. The exact mechanisms contributing to this effect are unknown however one possibility is that given that CPF and its active metabolite CPF oxon has been shown to disrupt microtubule networks this may be affecting intracellular localization[62]. Additional studies are needed to further investigate the mechanisms by which CPF reduces adrenergic signaling in BAT.

One of the limitations of our study is that we only examined the effects of CPF in male mice. It is known that CPF interacts with sex hormones[63] and steroid hormone receptors[64]. In terms

of the neurotoxic effects, males are more susceptible to organophosphates than females[65]. However, as reviewed[66], women may be more vulnerable to environmental pollution than men mainly because of the physiological differences caused by sex steroids. With a larger relative fat mass, women have a larger distribution volume for lipophilic substances than men[66]. Indeed, in aged human beings, concentrations of organochlorine pesticides are elevated in females[67]. Females may also have higher levels of thermogenic BAT than males[68]. Upon adrenergic activation, women have more potential to induce browning of perirenal adipose tissue than men due to sex-specific intrinsic factors[69] and have higher lipolytic activities[70]. Interestingly, females exhibit higher DIT than males at 20 °C but lower DIT than males at TN[71]. Future studies examining the effects of CPF in female mice are needed.

In conclusion, our study modeled the effects of an environmental toxicant under TN housing conditions. These studies have revealed that in addition to promoting food intake, CPF suppresses diet-induced thermogenesis in BAT and this exacerbates the development of obesity, NAFLD, and insulin resistance even at low doses which have no effect in mice housed at RT. These studies suggest that the effects of environmental toxicants on the development of obesity may have been underestimated as all studies to date have been conducted in mice housed at RT. Future studies examining the mechanisms driving reductions in β-AR signaling and whether there are associations between BAT metabolic activity and CPF in humans will be important.

## Methods

### In-vitro analysis

*Immortalized brown adipocyte cell line generation.* Cell lines were created from either FVB/N mice (wildtype immortalized brown adipocytes) or UCP-1-Luc2-TdTomato reporter mouse (immortalized UCP-1 reporter brown adipocytes), also known as the "ThermoMouse"[32] using reported techniques[72]. Briefly, BAT tissue was harvested from 4-day-old pups and subsequently digested with collagenase II, filtered, and plated. Isolated cells were then infected with a retrovirus created by transfecting Phoenix-ECO cells with a pBABE-SV40 plasmid from Addgene. Cells positive for SV40 were selected for 7 days using 2 µg/ml puromycin antibiotic and underwent a 14-day treatment with the antibiotic ciprofloxacin (10 µg/ml) in order to eliminate any mycoplasma contamination in the cells. UCP-1 Luc2-TdTomato (UCP-1 reporter line) was used for all luciferase assays to quantify UCP-1 promoter activity. For the cyclic AMP assay wildtype immortalized brown adipocytes were transfected with Promega pGloSensor-22F which can detect dynamic changes in cyclic AMP from concentration 0.003–10 µM in living cells.

*Immortalized murine brown adipocyte culture.* Immortalized brown adipocytes were cultured in high glucose DMEM (Gibco™) containing 10% fetal bovine serum and 1% antibiotic–antimycotic solution. For differentiation, confluent cells were placed in induction media (0.5 mM IBMX; 0.125 mM indomethacin; 0.2 µg ml$^{-1}$ dexamethasone; 1 nM T3, 20 nM insulin) for 2 days and subsequently maintained on differentiation media (1 nM T3, 20 nM insulin) for 7 days. For all cell experiments, cells were fully differentiated prior to compound or vehicle (≤0.1% dimethyl sulfoxide; DMSO) treatment.

*Cyclic AMP assay.* Cells were grown to confluence and fully differentiation as described above. Fully differentiated cells were seeded into 96-well assay plates at a 1:1 dilution based on well area. Following 48 h post-seeding, cells were treated with vehicle, CPF, or positive control propranolol for 7 days. Fresh media containing compound concentrations were added every 48 h. On day 7 a kinetic luciferase assay was performed. A 30 min basal read was first obtained, followed by a 100 nM isoproterenol injection and a 30 min stimulated kinetic read. Cyclic AMP activity is expressed as luminescence (RLU). Data are expressed as fold basal change (stimulated read (5–20 min)/basal read (5–30 min)).

*MitoTracker staining.* After washing with PBS, cells were treated with 200 nM MitoTracker Red and 100 nM MitoTracker Green staining solutions in DMEM (serum-free) for 25 min at 37 °C. Cells were then washed with PBS and stained with 1 µg/ml DAPI in DMEM. Images were taken using fluorescence microscopy and quantified using Image J.

*Cytochrome c oxidase activity.* Wildtype immortalized murine brown adipocytes were differentiated in 12-well plates. Cytochrome c oxidase activity was determined calorimetrically. The reaction mixture containing potassium phosphate buffer,

reduced cytochrome c, and ddH$_2$O, and cell lysate. Absorbance was immediately read at 550 nm wavelength for 3 min using SpectraMax M5 plate reader. The reaction was performed at 37 °C and cytochrome c oxidase activity was calculated as Eq. (1):

$$\text{Activity} = \frac{\triangle \text{OD/reaction time}}{\text{protein concentration} \times \text{sample volume}} \times \frac{\text{reaction volume}}{\text{exctinction coefficient} \times \text{path length}}$$

(1)

An extinction coefficient of 18.5 mmol$^{-1}$ cm$^{-1}$ (reduced cytochrome c), a path length of 0.625 cm, and a reaction volume of 0.0002 L were used. Protein was quantified during final calculations to determine the activity of enzyme per µg of protein. Each sample was run in triplicate on a 96-well plate.

*In vitro respiration assay of brown adipocytes.* The immortalized brown pre-adipocytes cell line were fully differentiated in a 75 cm$^2$ flask, then transferred to a 96-well flat-bottomed OxoPlate (OP96C, PreSens Precision Sensing) at a dilution of 1:8 by culture area. Before the assay, cell culture media was replaced with 200 µl respiration medium (minimal DMEM-D5030 supplemented with 10 mM HEPES, 25 mM glucose, 31 mM NaCl, 2 mM sodium pyruvate, 2 mL L-glutamine, and 4% BSA, pH = 7.4), covered by 75 µl heavy mineral oil and prewarmed at 37 °C in a SpectraMax M5 plate reader (Molecular Devices, San Jose, CA, USA) for 45 min. For calibration, each plate contained six wells of 0% O$_2$ standard (H$_2$O with 10 mg/mL sodium sulfite) or 100% O$_2$ standard (respiration media). The plate was kinetically read prior (basal) and following the addition of 2.5 µM oligomycin, 1 µM FCCP or 2 µM Rotenone +2 µM Antimycin A injections for 20 min. Oxygen levels were calculated according to the manufacturer's manual.

*RNA-sequencing analysis.* TMCON brown pre-adipocytes were differentiated in 24-well plates and treated with 1 pM CPF on day 9 of differentiation. Cells were treated for 4 h in Krebs–Ringer bicarbonate buffer (Sigma-Aldrich) supplemented with 10 mM HEPES (Janssen Pharmaceuticals, 3a2440 Geel, Belgium), 1 mM CaCl$_2$, and 1% fatty-acid free Bovine Serum Albumin (BSA) (EquiTech Bio Inc., TX, USA) at pH 7.4. Cell lysates were collected by administering 175 µL per well of RNeasy Lysis Buffer (RLT) (Qiagen, Hilden, Germany) supplemented with 143 mM beta-mercaptoethanol (Sigma-Aldrich) and vortexing until cells lifted from plates. Once cells were lifted from plates, 175 µL of 70% ethanol was added to each well, and lysates were transferred to a High Pure Filter tube from Roche RNA Isolation Kit (Roche Diagnostics, Mannheim, Germany). RNA isolation was done as outlined by manufacturers' specifications.

Raw sequencing reads (~12.5 million paired-end per sample; 50 base pairs each) were assessed for quality (such as GC content, PHRED scores, synthetic aptamer content, and sequence length) using FastQC (https://www.bioinformatics. babraham.ac.uk/projects/fastqc/). Trimmomatic[73] (default parameters) was used to trim low-quality reads, with the final set of reads aligned to the Ensembl *Mus musculus* (GRCm38.92) genome using HISAT2[74] (default parameters). The number of reads mapping to each gene was identified using htseq-count, with stranded set to "reverse", minimum alignment quality set to 10, feature type set to "exon" and "simple" advanced options. Mouse GTF gene annotation reference files were obtained from Ensembl. DESeq2[75] (default parameters) was used to normalize these data and a custom python script (McArthur) was used to convert the DESeq2 output into a ranked list of genes. The ranked list was used as input for Gene Set Enrichment Analysis (GSEA)[76,77], which was conducted using Cytoscape[78]. GSEA was performed using 1000 permutations, a false discovery rate cut-off of 25%, and a mouse gene set database from the Bader Lab at the University of Toronto (http://download.baderlab.org/EM_Genesets/current_release/Mouse/).

*Oil red O staining and quantification.* Cells were rinsed with PBS and fixed with 10% formalin for 40 min at RT. Fixed cells were incubated with Oil-Red O staining solution (Sigma Aldrich) for 10 min and then rinsed in 60% isopropanol to remove excessive Oil-Red O. Following imaging, Oil Red O stain was solubilized in isopropanol, and absorbance measures were taken at 510 nm utilizing Asys UVM340 microtiter plate reader.

*Radioligand ADRB3-binding competition assays.* Radioligand-binding competition activity was tested on the recombinant adrenergic β3 receptor for CPF and CPF oxon utilizing filtration binding assays by Epics and Euroscreen FAST (Belgium). Assays were performed in a 96-well plate (Master Block, Greiner, 786201), in duplicate. Binding buffer (25 mM Hepes pH 7.4, 1 mM EDTA, 0.5% BSA, 10 µg/ml saponin), membranes prepared from β3-receptor-expressing CHO-K1 cells (2 µg protein/well), radiotracer $^{125}$I-Cyanopindolol (0.165 nM final assay concentration) and test compound were utilized. A total of 9 concentrations were assessed per test compound between 0.0001 and 10,000 nM in duplicate. The assay was performed using reference competitor ZD 7114 (Tocris Pharmaceuticals) a β3 agonist. Non-specific binding was determined by co-incubation with 200-fold excess of a cold competitors. The samples were incubated in a final volume of 0.1 ml at 25 °C and for 30 min and then filtered over GF/B Unifilter plates presoaked for 2 h in 0.5% BSA. Filters are washed nine times with 0.5 ml of ice-cold washing buffer (25 mM Hepes pH 7.4, 0.5 M NaCl) and 50 µl of Microscint 20 (Packard) are added to each well. The plates were incubated for 15 min on an orbital shaker and then counted with a TopCount™ for 1 min/well. Dose–response data from test compounds

were analyzed with XLfit (IDBS) software using nonlinear regression applied to a sigmoidal dose–response model.

## In-vivo analysis

*Housing and diet.* All experiments were approved by the McMaster University Animal Ethics Committee and conducted under the Canadian guidelines for animal research. All mice used in this study were ordered at 7-week-old C57BL/6J males (JAX®, The Jackson Laboratory). They were housed in specific pathogen-free microisolator cages located in a room maintained at a constant temperature of 22 (room temperature, RT) or 30 °C (thermoneutrality, TN) on a 12-h light–dark cycle with lights on at 7:00 a.m., the humidity was 40–60%. The mice were fed a control diet (CD, 10 kcal% fat; D12450H research diet) or a high-fat diet (HFD, 45 kcal% fat; D12451 research diet) for 14 weeks. To avoid AChE inhibition[78] and mimic the level of environmental CPF exposure, the low dose CPF diet was supplemented with 2 mg/kg/DT (per kg diet) CPF for the first week; 3 mg/DT diet CPF (Toronto Research Chemicals, C425300, ON) for the second and third weeks and 5 mg/kg/DT CPF for 11 weeks, to provide 0.5 mg/kg/BW (per kg body weight) CPF. High dose CPF was 4 times the low dose. Based on food intake the actual CPF intake of each group was lower than the designed doses (Supplementary Fig. 2a). Diets and water were provided to the animal's ad libitum.

*Body composition measures.* Bodyweight (per mouse, $n = 10$) and body composition via Bruker's minispec Whole Body Composition Analyzer software and (Bruker-LF90-MRI) were monitored weekly.

*Metabolic testing.* Glucose- and insulin-tolerance tests (GTT and ITT) were performed after 12 and 13 weeks of treatment respectively. Mice were fasted for 6 h and intraperitoneally injected with glucose (1 g/kg in saline) or insulin (0.7 U/kg for CD and 1 U/kg for HFD mice). Blood samples were collected by tail vein bleed and analyzed using a glucometer (Accu-Chek®, Roche, 04680448003) immediately before and at 20, 40, 60, 90, and 120 min after injection of glucose or insulin.

Metabolic cages, Columbus Instruments Comprehensive Lab Animal Monitoring System (CLAMS), and Oxymax software (Columbus Instruments) were used to obtain $VO_2$, $VCO_2$, food intake, and activity data.

*Diet-induced thermogenesis measures.* Mice were fed with HFD (45 kcal% fat) supplemented with or without 20 mg/kg CPF for 5 weeks. Prior to 12 h of fasting, mice were placed in the CLAMS for 12 h for acclimatization and motoring the basal metabolic rate and activities from 7 p.m. to 7 a.m. Mice were then fasted from 7 a.m. to 7 p.m. and re-fed with 2 g of respective diet at 7 p.m. the following day.

*Tissue processing, histological examination, and transmission electron microscopy (TEM).* Tissues for histology were fixed in 10% neutral buffered formalin for 12 h at 4 °C, embedded in paraffin, and sectioned (5 µm thickness). Following deparaffinization, tissue sections were used for Hematoxylin-eosin (H&E) staining or immunostaining. The size of adipocytes was measured (three images per mouse) using Image J (NIH), representative images were selected according to the average adipocyte size. For immunostaining, sections were heated in citrate buffer for 20 min, blocked with 5% goat serum in TBS containing 0.3% Triton X-100 for 2 h, then incubated sequentially with primary antibodies overnight and secondary antibody for 1 h. The color was developed using a VECTOR® NovaRED™ Peroxidase (HRP) Substrate Kit (#SK-4800, Vector Laboratories). Sections were then mounted with a mounting medium (Vector Lab, Burlingame, CA). For liver used for lipid staining, tissues were infiltrated with 30% sucrose for 24 h after formalin fixation, then embed with OTC and frozen in isopentane cooled in liquid nitrogen. Cryo-sections of 10-µm-thick were rinsed with PBS, then stained with Oil Red O for 10 min, washed with 60% isopropanol for three times then mounted with an aqueous mounting media for further imaging.

Cell fractions of BAT tissue were isolated using a Membrane and Cytosol Protein Extraction kit according to the manufacturer's instructions (Beyotime, Cat. No. P0033, Shanghai, China).

For transmission electron microscopy (TEM), four interscapular BAT in each group were randomly selected and fixed in 2% glutaraldehyde (2% v/v) in 0.1 M sodium cacodylate buffer (pH 7.4) for 24 h. Tissues were sectioned on a Leica UCT ultramicrotome and picked up onto Cu grids. Sections were post-stained with uranyl acetate and lead citrate. The sectioning was performed by the electron microscopy group at McMaster University Medical Center. Mitochondria sizes was analyzed using Image J (NIH). The number of total mitochondria and mitochondria with disrupted cristae were counted. Cristae with any observable disorganization, vacuolization, or dissolution of cristae within mitochondria were categorized as disrupted. For each animal, more than 20 images were taken and analyzed. Researchers were blinded during image capturing and quantification. Representative images were selected according to the quantification data.

*Pair-feeding experiments.* Food intake of the control animals was measured every 3 days and the amount of food was provided to the pair-fed group in the following 3 days. For most of the time, the pair-fed animals treated with CPF ingested all the provided food. The difference in cumulative food intake of control and pair-fed mice was <0.1 g.

*AChE activity assay.* Acetylcholine esterase activity was measured in mouse skeletal muscle (hindlimb) and serum using Acetylcholinesterase Activity Assay Kit (Sigma-Aldrich Cat# MAK119). Skeletal muscle samples were homogenized in 0.1 M phosphate buffer (pH 7.5), centrifuged at 12,100×g for 5 min, and the supernatant was obtained. Diluted serum samples or supernatant from skeletal muscle samples was utilized as per the manufacturer's instruction. AChE activity was then normalized to protein content in the homogenates determined using the Pierce™ BCA Protein Assay Kit. The assay utilizes the Ellman method in which thiocholine reacts with 5,5-dithiobis (2-nitrobenzoic acid) to produce a colorimetric product indicative of AChE activity.

*Immunoblotting analysis.* Proteins were extracted from tissue or cells using cell lysis buffer (50 mM HEPES pH 7.4, 150 mM NaCl, 100 mM NaF, 10 Napyrophosphate, 5 EDTA mM, 250 mM sucrose, 1 mM DTT, and 1 mM Na-orthovanadate, 1% Triton X and Complete protease inhibitor cocktail (Roche)). Samples (1 µg/µL) were prepared in 4× SDS sample buffer and boiled at 95 °C for 5 min. Sample proteins were separated in a 10% or 12% SDS–PAGE gel and transferred to a PVDF membrane at 100 V for 90 min. Membranes were blocked with 5% skim milk in TBST (50 mM Tris–HCl, 150 mM NaCl, 0.05 % Tween 20) for 1 h. Subsequently, membranes were subject to primary antibody (1:1000 dilution) in TBST (with 5% BSA) overnight at 4 °C and then incubated with secondary antibody (1:1000 dilution) at room temperature for 1 h. Protein bands were imaged using electrochemiluminescence and analyzed using Image J software (National Institutes of Health, Bethesda, USA). Antibodies against pACC$^{S79}$ (#3661), ACC (#3662), pAMPKα$^{T172}$ (#2535), AMPKα (#2532), pULK1$^{S555}$ (#5869), ULK1 (#8054), p-HSL$^{S660}$ (#4126), HSL (#4107), LC3B (#2775), p38 (#8690), p-p38 (#9211), p62 (#5114), pPKA substrate (#9621) and p62 (#5114) were purchased from Cell Signaling (Danvers, MA). Antibody against UCP1 (#UCP11-A) was purchased from Alpha Diagnostic International. Antibody against β-tubulin (#32-2600) was purchased from Invitrogen.

*Tissue and serum triglyceride determination.* Approximately 50 mg of tissue was homogenized in chloroform:methanol (2:1) and a portion of the organic phase was dried down and suspended in isopropanol. Serum or tissue samples were assayed for triglyceride amount using a Triglyceride Colorimetric Assay Kit (Cayman Chemical, MI).

*RNA isolation and quantitative real-time PCR (qRT-PCR).* Total RNA was isolated using TRIzol (Life Technologies, Grand Island, NY, USA) and purified using an RNeasy kit (QIAGEN) column. cDNA was synthesized using iScript™ cDNA Synthesis Kit (Bio-Rad, Hercules, CA). qRT-PCR was performed in a qPCR thermocycler (Corbett Rotor Gene 6000, MBI, QC, Canada) using TaqMan primers purchased from Invitrogen (Supplementary Table 1). Relative gene expression was calculated using the comparative Ct ($2^{-\Delta\Delta Ct}$) method, where values were normalized to a housekeeping gene (*Ppia*).

## In-silico analysis

*Docking of CPF to $\beta_3$-AR.* To predict the binding pocket and binding mode of CPF in human beta-3 adrenergic receptors ($\beta_3$-AR), we employed an approach combining molecular dynamics (MD) simulations and docking calculations. Since the experimental 3-D structure of $\beta_3$-AR is unavailable, we used the multiple sequence alignment approach to build a homology model. a $\beta_3$-AR homology model was used as a starting structure to carry out classical MD simulation. The MD simulations were performed using GROMACS v5.1.2[79]. AutoDock Vina[80] was then used to dock CPF to the 100 snapshots each of $\beta_3$AR extracted from the MD simulation to determine the pocket and high scoring pose.

*$\beta_3$-AR multiple sequence alignment and model building.* The amino acid sequence of $\beta_3$-AR was downloaded from UniProt (http://www.uniprot.org/) using accession number P13945. A template structure search was first carried out using SWISS-MODEL (https://swissmodel.expasy.org/). The template structures with a sequence identity > 50% and high GMQE score was selected for building the three-dimensional (3-D) structure of $\beta_3$-AR. The selected template structures were chain B of activated turkey $\beta_1$-AR (56.33%; PDB ID: 6H7J)[81] and chain A of human $\beta_2$-AR (50%; PDB ID: 3NY8)[82]. MODELLER v9.20[83] was then used to perform multiple sequence alignment and generate the 3-D structure of $\beta_3$-AR. Prior to multiple sequence alignment, the G protein-like antibody, T4-lysozyme, seen in the template structure of human $\beta_2$-AR was removed. Five 3-D structures of $\beta_3$-AR were generated. Stereochemical checks were performed using the SWISS-MODEL structure assessment tool (https://swissmodel.expasy.org/) on all five structures and the best was chosen based on minimal stereochemical deviation and low Molprobity score. The final selected model had 96.74% of amino acid residues in the Ramachandran favored region with no outlier. Molprobity score, which combines the clash score, rotamer, and Ramachandran evaluations into a single score, normalized to be on the same scale as X-ray resolution for the model was 2.80. root mean square deviation (RMSD) between backbone atoms of final $\beta_3$-AR model and $\beta_2$-AR experimental structure was 3.6 Å. The final modeled structure of $\beta_3$-AR lacked N-terminal amino acid residues 1–32, intracellular loop 237–284, and C-terminal amino acids 361–408.

*System setup and molecular dynamics simulation.* The homology model of $\beta_3$-AR was used as a starting structure to carry out MD simulation. The structure file was first uploaded on the web server, Prediction of Proteins in Membranes (http://opm.phar.umich.edu/server.php). Membrane boundaries provided by this server along with the protein model were then uploaded onto the CHARMM-GUI server (http://charmm-gui.org/) for further processing. Protonation states of amino acid residues were assigned at the physiological pH of 7.4. The protein was embedded in 1-palmitoyl-2-oleoyl-sn-glycero-3-phosphocholine. The replacement method was used to pack the receptor within the lipid bilayer. The lipid layer thickness was chosen to be 1.6 (80 lipids in a top leaflet and 78 lipids in the bottom leaflet). This system was then placed in a rectangular solvent box, and a 15 Å TIP3P water layer was added to solvate intra-and extra-cellular space. Charge neutrality of the system was achieved by adding Na+ and Cl− ions at a concentration of 0.15 mol/L to the water layers. 10 ns MD simulation of this system was then performed using the protocol described in Patel et al.[84]. During the 10 ns production simulation, snapshots were saved every 100 ps giving 100 snapshots of $\beta_3$-AR to be used for docking calculations.

*Docking of CPF to $\beta_3$-AR.* Each of the 100 snapshots obtained during MD simulations was used to dock CPF using AutoDock Vina[80]. The docking program allows the ligand to be completely flexible during the conformational search. However, only a few restricted side chains on the protein can be assigned as flexible. The use of 100 snapshots from MD simulation allows us to at least partly overcome this limitation.

CPF was sketched in 2-D using the ChemDraw tool of the ChemBio office package (http://www.cambridgesoft.com). The 2-D structure was converted to 3-D using the Chem3D tool. Ligand geometry was then optimized by performing an energy minimization run using MMFF94force-field. Lastly, the optimized 3-D structure of CPF was converted to the PDBQT format using AutoDockTools (http://mgltools.scripps.edu/) with the default identification of rotatable bonds and Gasteiger partial charges.

Ligand docking was then carried out by creating a grid box of size 28 Å × 28 Å × 28 Å, centered at a known ligand-binding site, with a grid spacing of 1 Å. This grid was large enough to allow exploration of all possible CPF-binding orientations surrounding known AR-binding pocket in an unbiased manner. The input exhaustiveness parameter for the docking was set to 500. The number of top docking orientations with high docking scores was fixed to 15. This docking protocol was similarly applied to all 100 snapshots of $\beta_3$-AR. CPF docking pose with the highest docking score was extracted and analyzed.

*Statistical analysis.* All data were found to be normally distributed. Results were analyzed using Student's *t*-test or ANOVA where appropriate, using GraphPad Prism software. A repeated-measures ANOVA was used for all bodyweight plots, fed blood glucose, and GTT and ITT data. A Bonferroni post hoc test was used to test for significant differences revealed by the ANOVA. Significance was accepted at $p \leq 0.05$.

**Reporting summary**. Further information on research design is available in the Nature Research Reporting Summary linked to this article.

## Data availability

RNA-sequencing data are deposited in Gene Expression Omnibus (GEO) under accession number GSE178366. All data supporting the findings of this study are provided within the paper and its supplementary information. All additional information will be made available upon reasonable request to the corresponding author. Source data are provided with this paper.

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

## Acknowledgements

The funding for this study was provided by the Canadian Institutes of Health Research (CIHR) Programmatic Grants in Environment, Genes and Chronic Diseases (to A.G.M., M.G.W., K.M.M., A.C.H. and G.R.S.—Grant #144625-1). G.R.S. is supported by a Canada Research Chair and a J. Bruce Duncan Chair in Metabolic Diseases. E.M.D. is a Vanier Scholar. J.S.P. was supported by the Center for Modeling Complex Interactions sponsored by the NIGMS under award number P20 GM104420. Computational resources were provided by the Institute for Bioinformatics and Evolutionary Studies Computational Resources Core sponsored by the National Institutes of Health (NIH P30 GM103324) and by the High Performance Computing Center at Idaho National Laboratory, which is supported by the Office of Nuclear Energy of the U.S. DOE and the Nuclear Science User Facilities under Contract No. DE-AC07-05ID14517.

## Author contributions

B.W., S.Z., A.L., E.T., E.M.D., J.Y., A.G., E.D., B.K., J.L., J.P., J.W. performed the experiments. B.W., E.M.D., A.G., E.D., J.P., A.R., K.S., A.M., S.K., M.W., K.M., A.H. and G.R.S. provided intellectual input. B.W., S.Z., A.L. and G.R.S. wrote the manuscript with contributions from all authors. E.M.D., B.W. and E.T. created the graphical abstract.

## Competing interests

The authors declare no competing interests.
