## [Peer Review File · Nature Communications]

Reviewer comments, first round of review:

Reviewer #1 (Remarks to the Author):

Wang et al. have assembled a comprehensive manuscript examining the role of chlorpyrifos in disrupting thermogenesis and promoting global metabolic dysfunction. The studies use a complementary array of techniques to examine the mechanisms by which chlorpyrifos alters brown adipose function and metabolic regulation from the perspective of the intact organism down to molecular interactions between chlorpyrifos and adrenergic receptors. The quality and depth of the studies cannot be overstated. The data presented in the manuscript and supplementary materials clearly demonstrate chlorpyrifos's ability to disrupt metabolic function, raising important questions about its importance to human health.

This point of human health relevance is reinforced by the fact that chlorpyrifos is an important environmental toxicant that has been thrust into regulatory purgatory in the U.S. As such, these data are vital and timely additions to the armamentarium of studies that argue for policy interventions to protect human health. The value of this data stem from the thoroughness of the studies and the wide spectrum of analyses supporting the findings. Moreover, examination of effects at low-dose exposures is of special relevance to human health. Overall, this manuscript was a delight to read.

My reservations about the manuscript are relatively minor, but include:

1. The use of only male mice for the in vivo studies is a limitation that should be acknowledged. Future studies should most definitely examine sex-specific effects, especially given suggestions that chlorpyrifos can disrupt sex steroid hormones and can exert sex-specific neurotoxic effects.
2. Acknowledging space limitations, it would still be useful to acknowledge other studies that have shown disruptions of thermogenesis by environmental toxicants (e.g. studies by the La Merrill group examining arsenic).
3. Other studies have shown metabolic disruptions induced by chlorpyrifos in a diet-specific fashion that have been linked to changes in the microbiome. It would be useful to acknowledge these studies and the possibility that chlorpyrifos may be acting via multiple mechanisms (that admittedly may be concentration dependent). I do, however, think that the authors' manuscript clearly argues for very clear linkage to impaired diet-induced thermogenesis.
4. Linkage to epidemiological studies examining chlorpyrifos's association with metabolic dysfunction in humans would be helpful. And if such studies do not exist, that is also of importance to know.
5. Very minor, but please define the abbreviation TN when first used in the manuscript. Again, very minor.

In conclusion, this is really excellent work that was a delight to read...and will hopefully support efforts to regulate chlorpyrifos in order to promote human health.

Wishing you and your families safety and health during these challenging time.

Reviewer #2 (Remarks to the Author):

Wang and colleague investigated the effects of several environmental toxins and pesticides, which are generally associated with food, on brown adipose tissue function. The authors identified the pesticide chlorpyrifos as a suppressor of UCP1 expression in brown adipocytes in vitro and vivo. Suppression of brown fat leads to increased weight gain and obesity-associated disorders such as fatty liver and insulin resistance under thermoneutral conditions. Furthermore, the authors claim that the suppression of thermogenesis by chlorpyrifos (CPF) is associated with impaired activation

of adrenergic β 3 receptors.

While it is known that CPF uptake induces body weight gain, the impact on energy metabolism, however, in particular on diet-induced thermogenesis, has not been investigated before. In total, the experimental studies of the paper are very well performed and the results are interesting for the readership working on metabolic diseases.

I have one major concern:

The modeling of CPF binding to β adrenergic receptors suggests that CPF directly inhibits β -adrenergic receptors. However, CPF treatment causes increased fat gain only when mice were kept on a high fat diet at 30°C, when no additional heat production/activation of β adrenergic receptors is necessary for thermoregulation, but not at 21°C. This observation is unexpected, especially as the transcription of UCP1 in vitro is repressed immediately (4h treatment) and CPF reduced cAMP during isoproterenol stimulation. Furthermore the results are in contrast to cit. 38 (doi:10.1016/j.envres.2015.06.036), where CPF treatment leads to increased body weight gain even at room temperature. To strengthen their findings, the authors may want to combine CPF treatment with pharmacological β adrenergic activation in vivo.

Further concerns:

1. Most of the genes listed in table S1 are up-regulated upon CPF treatment, which suggests transcriptional activation but not repression of mitochondrial function after 4 hours CPF treatment. Indeed, the authors show that 4 hours of treatment do not lead to reduced maximal respiration (Fig S1d), which would be seen after 6 days of treatment. This discrepancy of the results on mitochondrial function between acute and chronic treatment with CPF is quite interesting. Here, the conclusion would be strengthened by analyses of RNA seq of 6 days-treated cells compared to 4h treatments.
2. For protein and functional assays, the cells were treated for 6d with CPF. Does this treatment affect differentiation of the cells? Fat cells shift from more glycolytic (preadipocytes) to more oxidative (mature adipocytes) metabolism during differentiation. Furthermore UCP1 is induced quite late during differentiation. Thus, it would be important to demonstrate that the treatment does not inhibit/repress differentiation.
3. How long were the animals treated with HFD before indirect calorimetry? Was a difference in body weights between the groups already observed? The energy metabolism studies are key to identify the mechanisms underlying the obesity progression in CPF treated animals. Due to the high animal number, it would be helpful for data interpretation if the authors would also show resting and daily O₂ consumption, plotted against the body weight. Furthermore a separation of day/night energy expenditure could be of interest. Is the effect primarily seen during night when mice are eating (due to diet-induced thermogenesis)?
4. CPF suppressing DIT during the fasting/refeeding experiment of animals with identical body weight is a convincing observation. Was this experiment also performed for HFD fed mice at RT?

Reviewer #3 (Remarks to the Author):

I am an expert in GPCR structural biology and therefore I will limit my comments on the manuscript to a single aspect, namely whether the beta3-adrenoceptor is the target for CPF. There are two aspects of the manuscript that give me concern, the docking studies of CPF on a model of beta3AR and the extraordinary potency of CPF on inhibiting thermogenesis.

Docking of known ligands into structures of GPCRs can often result in ligand poses that bear no resemblance to the ligand pose determined by X-ray crystallography or cryo-EM. This of course will be exacerbated by performing docking in a model. Given the extensive work on beta receptors and their high homology I would think that the model for beta3AR could be good, although no data is shown in the manuscript for this. Therefore, although CPF appears to dock into beta3AR and beta2AR, subsequent confirmation must be sought through experimental means. The extremely high potency of CPF in inhibiting thermogenesis (1 pM) suggests to me that it might even be a covalent modification of a protein(s) that causes the observed in vivo effects. There is nothing in

the model shown of CPF docked into beta3AR that would suggest such a high affinity of binding. Even though potency and affinity can differ at a given GPCR, I would have expected to see a dramatic difference between CPF and the binding poses of other beta receptor ligands that bind typically with nM affinity.

Both of the above issues mean that it is imperative to perform additional experiments to show that beta3AR is the target for CPF. I have made a couple of suggestions below.

1. Perform radioligand competition binding curves, where the binding of a known radioligand of beta3AR (e.g. iodocyanopindolol) is inhibited by increasing concentrations of CPF. If CPF binds to beta3AR, then there should be potent (sub-nM) inhibition of binding of another ligand. There are many known ligands of beta3AR which could be used as positive controls (see the IUPHAR GPCR database).
2. In Figure 4, multiple panels show the decrease in phosphorylation of proteins downstream of beta3AR in the signalling pathway after treatment with CPF in brown adipocytes. If inhibiting beta3AR is all that is required for this effect, then adding an inverse agonist of beta3AR (e.g. ICI118551) should have the identical effect as CPF in inhibiting thermogenesis. In addition, adding a partial agonist of beta3AR would be expected to inhibit the effect of CPF.
3. It would be helpful to cite evidence of the abundance and type of beta receptors found at the cell surface of brown adipocytes.
4. Please state the RMSD between the beta3AR model and beta2AR structure. You could also perform a docking of cyanopindolol to the model and compare it to crystal structures of b1AR bound to cyanopindolol, which I would expect to be very similar.

REVIEWER COMMENTS

Reviewer #1 (Remarks to the Author):

Wang et al. have assembled a comprehensive manuscript examining the role of chlorpyrifos in disrupting thermogenesis and promoting global metabolic dysfunction. The studies use a complementary array of techniques to examine the mechanisms by which chlorpyrifos alters brown adipose function and metabolic regulation from the perspective of the intact organism down to molecular interactions between chlorpyrifos and adrenergic receptors. The quality and depth of the studies cannot be overstated. The data presented in the manuscript and supplementary materials clearly demonstrate chlorpyrifos's ability to disrupt metabolic function, raising important questions about its importance to human health.

This point of human health relevance is reinforced by the fact that chlorpyrifos is an important environmental toxicant that has been thrust into regulatory purgatory in the U.S. As such, these data are vital and timely additions to the armamentarium of studies that argue for policy interventions to protect human health. The value of this data stem from the thoroughness of the studies and the wide spectrum of analyses supporting the findings. Moreover, examination of effects at low-dose exposures is of special relevance to human health. Overall, this manuscript was a delight to read.

Thank-you for the very positive comments.

My reservations about the manuscript are relatively minor, but include:

1. The use of only male mice for the in vivo studies is a limitation that should be acknowledged. Future studies should most definitely examine sex-specific effects, especially given suggestions that chlorpyrifos can disrupt sex steroid hormones and can exert sex-specific neurotoxic effects.

Response: Thank you for this suggestion. We have inserted 2 sentences in the discussion indicating that this is one limitation of the study and that this should be addressed in future research.

2. Acknowledging space limitations, it would still be useful to acknowledge other studies that have shown disruptions of thermogenesis by environmental toxicants (e.g. studies by the La Merrill group examining arsenic).

Response: These and other important studies related to disruptions in thermogenesis by environmental toxicants have been acknowledged in the introduction as suggested (line 61-62).

3. Other studies have shown metabolic disruptions induced by chlorpyrifos in a diet-specific fashion that have been linked to changes in the microbiome. It would be useful to acknowledge these studies and the possibility that chlorpyrifos may be acting via multiple mechanisms (that admittedly may be concentration dependent). I do, however,

think that the authors' manuscript clearly argues for very clear linkage to impaired diet-induced thermogenesis.

Response: We have now cited the multiple studies which have indicated effects of chlorpyrifos to promote obesity in animal models and indicated this may involve increase in appetite and alterations in the gut microbiome. (line 89-90)

4. Linkage to epidemiological studies examining chlorpyrifos's association with metabolic dysfunction in humans would be helpful. And if such studies do not exist, that is also of importance to know.

Response: To the best of our knowledge there have been no epidemiological studies linking chlorpyrifos with obesity in humans and this has now been stated (line 91-92)

5. Very minor, but please define the abbreviation TN when first used in the manuscript. Again, very minor.

Response: Revised, thank you. (line 122)

In conclusion, this is really excellent work that was a delight to read...and will hopefully support efforts to regulate chlorpyrifos in order to promote human health.

Wishing you and your families safety and health during these challenging time.

Response: Thank you again for your comments and kind wishes.

Reviewer #2 (Remarks to the Author):

Wang and colleague investigated the effects of several environmental toxins and pesticides, which are generally associated with food, on brown adipose tissue function. The authors identified the pesticide chlorpyrifos as a suppressor of UCP1 expression in brown adipocytes in vitro and vivo. Suppression of brown fat leads to increased weight gain and obesity-associated disorders such as fatty liver and insulin resistance under thermoneutral conditions. Furthermore, the authors claim that the suppression of thermogenesis by chlorpyrifos (CPF) is associated with impaired activation of adrenergic β 3 receptors.

While it is known that CPF uptake induces body weight gain, the impact on energy metabolism, however, in particular on diet-induced thermogenesis, has not been investigated before. In total, the experimental studies of the paper are very well performed and the results are interesting for the readership working on metabolic diseases.

I have one major concern:

The modeling of CPF binding to β adrenergic receptors suggests that CPF directly inhibits

β -adrenergic receptors. However, CPF treatment causes increased fat gain only when mice were kept on a high fat diet at 30°C, when no additional heat production/activation of β adrenergic receptors is necessary for thermoregulation, but not at 21°C. This observation is unexpected, especially as the transcription of UCP1 in vitro is repressed immediately (4h treatment) and CPF reduced cAMP during isoproterenol stimulation.

Response: It is important to note that we did detect increased food intake (Fig 3A), weight gain, glucose intolerance and insulin resistance in mice treated with our highest dose of CPF at room temperature (Fig S4). However, when mice were housed at thermoneutrality both low and high doses of CPF had more dramatic effects on weight gain and glucose intolerance/insulin resistance (Fig 3).

We do not believe this effect is unexpected given the proposed mechanism of regulating diet-induced thermogenesis. This is because housing mice at room temperature increases basal metabolic rate due to adaptive thermogenesis and importantly this adaptive thermogenesis can occur in mice lacking beta-adrenergic receptors or UCP1 due to upregulation of many distinct compensatory pathways (e.g. shivering, creatine cycling, g-beige fat). However, this does not mean that b-adrenergic signaling and UCP1 are not important for regulating diet-induced thermogenesis but rather animals are able to adapt via alternative pathways in order to survive.

Therefore, our data indicating that the blunting of the obesity promoting effect at room temperature is likely due to upregulation of adaptive thermogenesis pathways which are not dependent on b-adrenergic and/or UCP1, findings what are entirely consistent with observations in UCP1 null mice.

We have attempted to more clearly rationalize the importance of the TN housing as it related to adaptive and diet-induced thermogenesis in greater detail in lines 126-136 in the revised paper.

Furthermore the results are in contrast to cit. 38 (doi:10.1016/j.envres.2015.06.036), where CPF treatment leads to increased body weight gain even at room temperature. To strengthen their findings, the authors may want to combine CPF treatment with pharmacological β adrenergic activation in vivo.

Response: As stated above we did see weight gain in mice at room temperature when fed the highest dose of CPF (2.0mg/kg). In the above cited paper C57Bl6/N and ApoeC3 mice were housed a room temperature and treated with CPF at a dose of 2.0mg/kg body weight. These authors found that this dosing increased food intake, weight gain, blood glucose and insulin resistance in ApoeC3 mice and that this effect was also observed in C57Bl6/N mice but to a lesser degree. Therefore, our data is entirely consistent with their findings (Supplementary Figure S4A-D).

Further concerns:

1. Most of the genes listed in table S1 are up-regulated upon CPF treatment, which suggests transcriptional activation but not repression of mitochondrial function after 4 hours CPF treatment. Indeed, the authors show that 4 hours of treatment do not lead to reduced maximal respiration (Fig S1d), which would be seen after 6 days of treatment. This discrepancy of the results on mitochondrial function between acute and chronic treatment with CPF is quite interesting. Here, the conclusion would be strengthened by analyses of RNA seq of 6 days-treated cells compared to 4h treatments.

Response: We have reanalyzed our RNAseq dataset using more up to date computational tools and these new data are presented in Fig 1d-e. As the reviewer can now see the number of mitochondrial associated transcripts which are upregulated and downregulated are equally represented.

2. For protein and functional assays, the cells were treated for 6d with CPF. Does this treatment affect differentiation of the cells? Fat cells shift from more glycolytic (preadipocytes) to more oxidative (mature adipocytes) metabolism during differentiation. Furthermore UCP1 is induced quite late during differentiation. Thus, it would be important to demonstrate that the treatment does not inhibit/repress differentiation.

Response: We agree it is very important to clarify whether the effects of CPF on UCP1 and mitochondrial function are secondary to transcriptional repression of BAT differentiation. We now provide two lines of evidence which suggests this is not the case. These data are (lines 108-110): There were no impairments in markers of BAT differentiation as Oil Red O staining and key transcription factors/markers of BAT differentiation (*Prdm16*, *Ppargc1a*, *Pparg*) were not altered by CPF (Supplementary Figure 1d, e).

3. How long were the animals treated with HFD before indirect calorimetry? Was a difference in body weights between the groups already observed? The energy metabolism studies are key to identify the mechanisms underlying the obesity progression in CPF treated animals. Due to the high animal number, it would be helpful for data interpretation if the authors would also show resting and daily O₂ consumption, plotted against the body weight. Furthermore, a separation of day/night energy expenditure could be of interest. Is the effect primarily seen during night when mice are eating (due to diet-induced thermogenesis)?

Response: For chronic CPF treatment (14 weeks), mice had been treated with CPF for 11 weeks when applied for metabolic analysis. For these mice, there was a difference in body weight between control and CPF treated groups (Figure 2a). However, it is important to note that we plotted O₂ consumption in figure 3c NOT corrected for body mass. And even despite having a greater body mass O₂ consumption was still lower in CPF treated mice. Therefore correcting for body mass would make this difference even larger however, we have not done this as this analysis has been suggested to be avoided ¹.

With respect to day and night variability we have plotted this for the reviewer below. As the reviewer can see there are reductions in O₂ consumption during both day and night a finding that is consistent with the food intake data which shows that mice ate during both light and dark cycles. As previously described² this lack of circadian rhythm with respect to feeding is a hallmark of HFD-fed mice. As such we believe including these data in the main manuscript will not be beneficial especially in light of our much better controlled experiment provided in Figure 4 in which diet-induced thermogenesis was measured by fasting and then refeeding within the metabolic cages before appreciable changes in body mass occurred (now Figure 5)

4. CPF suppressing DIT during the fasting/refeeding experiment of animals with identical body weight is a convincing observation. Was this experiment also performed for HFD fed mice at RT?

Response: We did not complete this experiment in mice housed at RT. As discussed in response 1 above previous studies have shown that detecting increases in DIT is difficult in mice housed at RT.

Reviewer #3 (Remarks to the Author):

I am an expert in GPCR structural biology and therefore I will limit my comments on the manuscript to a single aspect, namely whether the beta3-adrenoceptor is the target for CPF. There are two aspects of the manuscript that give me concern, the docking studies of CPF on a model of beta3AR and the extraordinary potency of CPF on inhibiting thermogenesis.

Docking of known ligands into structures of GPCRs can often result in ligand poses that bear no resemblance to the ligand pose determined by X-ray crystallography or cryo-EM. This of course will be exacerbated by performing docking in a model. Given the extensive work on beta receptors and their high homology I would think that the model for beta3AR could be good, although no data is shown in the manuscript for this. Therefore, although CPF appears to dock into beta3AR and beta2AR, subsequent confirmation must be

sought through experimental means. The extremely high potency of CPF in inhibiting thermogenesis (1 pM) suggests to me that it might even be a covalent modification of a protein(s) that causes the observed in vivo effects. There is nothing in the model shown of CPF docked into beta3AR that would suggest such a high affinity of binding. Even though potency and affinity can differ at a given GPCR, I would have expected to see a dramatic difference between CPF and the binding poses of other beta receptor ligands that bind typically with nM affinity.

Response:

We thank the reviewer for their careful analysis of our model and the suggestion that further experimental evidence was required to support that this was the primary mechanism of action by which CPF-inhibits cAMP.

Quality of β_3 -AR homology model was assessed using SWISS-MODEL structure assessment tool. Final selected model had 96.74% of amino acid residues in the Ramachandran favored region with no outlier. Molprobit score, which combines the clashscore, rotamer, and Ramachandran evaluations into a single score, normalized to be on the same scale as X-ray resolution for the model was 2.80. We also repeated our docking calculations using the same protocol but with different random seed to ensure we are not missing out the best docking solution of CPF due to insufficient of sampling. This re-docking exercise did not yield a better docked pose in the β_3 -AR model than the ones which were shown in the earlier version of the manuscript.

Both of the above issues mean that it is imperative to perform additional experiments to show that beta3AR is the target for CPF. I have made a couple of suggestions below.

1. Perform radioligand competition binding curves, where the binding of a known radioligand of beta3AR (e.g. iodocyanopindolol) is inhibited by increasing concentrations of CPF. If CPF binds to beta3AR, then there should be potent (sub-nM) inhibition of binding of another ligand. There are many known ligands of beta3AR which could be used as positive controls (see the IUPHAR GPCR database).

Response: As suggested radioligand competition binding assays with CPF and CPF-oxon with iodocyanopindolol were performed by EuroScreenFast. As is now shown presented in Supplementary Figure 9b and below these radioligand competition binding curves suggest CPF nor CPF-oxon bind directly to the b3AR. ZD7114 a known b3AR agonist was utilized as a positive control.

2. In Figure 4, multiple panels show the decrease in phosphorylation of proteins downstream of beta3AR in the signaling pathway after treatment with CPF in brown adipocytes. If inhibiting beta3AR is all that is required for this effect, then adding an inverse agonist of beta3AR (e.g. ICI118551) should have the identical effect as CPF in inhibiting thermogenesis. In addition, adding a partial agonist of beta3AR would be expected to inhibit the effect of CPF.

Response: We have treated BAT cells with b3AR antagonist propranolol for 72h followed by a 3min isoproterenol treatment at 10nM. Cell lysates were prepared, and WB analysis was performed as per the experiments with CPF. These data presented in Supplementary Figure 10a and below show that CPF treatment mirrors that of b3AR antagonist, decreasing phosphorylation of HSL downstream of the beta3AR

3. It would be helpful to cite evidence of the abundance and type of beta receptors found at the cell surface of brown adipocytes.

Response: As now presented in the manuscript in Supplementary Figure 9 and below the B3AR represents the most abundant of the beta receptors in brown adipocytes of mice. These findings are in agreement with previous studies in the literature which have also shown that the B3AR is the main receptor in mice while the B2AR is the primary receptor in human BAT³.

4. Please state the RMSD between the beta3AR model and beta2AR structure. You could also perform a docking of cyanopindolol to the model and compare it to crystal structures of b1AR bound to cyanopindolol, which I would expect to be very similar.

Response: Root Mean Square Deviation (RMSD) between β_3 -AR model and β_2 -AR experimental structure is 3.6 Ang. As suggested, we validated our β_3 -AR model by performing a docking of cyanopindolol to 100 snapshots extracted from 10 ns long molecular dynamics simulations. Top docking pose was compared against the co-crystal of β_1 -AR bound to cyanopindolol by superimposing C-alpha atoms of β_3 -AR model and β_1 -AR structure. We observed a similar binding mode for cyanopindolol in both the structures, which demonstrates the accuracy of our β_3 -AR model.

Fig. Comparison of binding mode of cyanopindolol in β_3 -AR homology model and β_1 -AR experimental structure. β_1 -AR structure is represented using gray cartoon. Experimental binding conformation (purple carbon atoms) and docked pose (orange carbon atoms) of cyanopindolol are shown using stick representation. Amino acid residues (gray sticks) involved in hydrogen bond interaction (black dashed lines) with cyanopindolol in β_1 -AR structure are shown in an enlarged view. Nitrogen and Oxygen atoms are shown using blue and red colors respectively.

References

- 1 Tschöp, M. H. *et al.* A guide to analysis of mouse energy metabolism. *Nature methods* **9**, 57-63 (2012).
- 2 Hatori, M. *et al.* Time-restricted feeding without reducing caloric intake prevents metabolic diseases in mice fed a high-fat diet. *Cell metabolism* **15**, 848-860 (2012).
- 3 Blondin, D. P. *et al.* Human brown adipocyte thermogenesis is driven by β 2-AR stimulation. *Cell Metabolism* **32**, 287-300. e287 (2020).

Reviewer comments, second round of review:

Reviewer #1 (Remarks to the Author):

The authors have addressed all of my concerns. Furthermore, the additional data provided in response to the other reviewers further strengthen the manuscript. I have no further concerns and wish everyone health and safety during these challenging times.

Reviewer #2 (Remarks to the Author):

No additional comments.

Reviewer #3 (Remarks to the Author):

The authors have performed the additional experiments requested on b3AR that show there is no direct binding of CPF or oxon-CPF to the receptor. In the light of this, they now state that the molecular mechanism of CPF activity is unclear. I think that this is a fair statement, and I have no further comments on the manuscript.

REVIEWERS' COMMENTS

Reviewer #1 (Remarks to the Author):

The authors have addressed all of my concerns. Furthermore, the additional data provided in response to the other reviewers further strengthen the manuscript. I have no further concerns and wish everyone health and safety during these challenging times.

Response: Thank you.

Reviewer #2 (Remarks to the Author):

No additional comments.

Response: Thank you.

Reviewer #3 (Remarks to the Author):

The authors have performed the additional experiments requested on b3AR that show there is no direct binding of CPF or oxon-CPF to the receptor. In the light of this, they now state that the molecular mechanism of CPF activity is unclear. I think that this is a fair statement, and I have no further comments on the manuscript.

Response: Thank you.